# Dengue seroprevalence study in Bali

**Sri Masyeni**[1]*, **Rois Muqsith Fatawy**[2], **A. A. A. L. Paramasatiari**[3],
**Ananda Maheraditya**[3], **Ratna Kartika Dewi**[3], **N. W. Winianti**[3], **Agus Santosa**[3],
**Marta Setiabudy**[3], **Nyoman Trisna Sumadewi**[4], **Sianny Herawati**[4]

1 Faculty of Medicine and Health Science, Department of Internal Medicine, University of Warmadewa, Bali,
Indonesia, 2 Faculty of Medicine, Infectious Disease and Immunology Research Center, Indonesia Medical
Education and Research Institute, Universitas Indonesia, Jakarta, Indonesia, 3 Faculty of Medicine and
Health Science, University of Warmadewa, Bali, Indonesia, 4 Faculty of Medicine, University of Udayana,
Bali, Indonesia

* masyeniputu@yahoo.com

## Abstract

### Introduction

Dengue infection poses significant public health problems in tropical and subtropical regions worldwide. The clinical manifestations of dengue vary from asymptomatic to severe dengue manifestations. This serological survey highlighted the high incidence of asymptomatic cases. This study aimed to determine the prevalence of dengue in healthy and ill adults in Bali.

### Methods

Cross-sectional seroprevalence surveys were performed between July 2020 and June 2021 among healthy and ill adults in Denpasar Bali. Blood samples were collected from 539 randomly selected urban sites in Denpasar. Immunoglobulin G antibodies against the dengue virus were detected in serum using a commercial enzyme-linked immunosorbent assay kit.

### Results

Overall, the dengue seroprevalence rate among the 539 clinically healthy and ill adults was high (85.5%). The median age was 34.1 (18–86.1). Most of the participants in the study were younger than 40 years (61.2%). Men were the dominant sex (54.5%). The study found a significant association between dengue seropositivity among people aged > 40 years and healthy status (p = 0.005; odds ratio [OR] = 0.459 and p < 0.001; OR = 0.336, respectively). The study reported that as many as 60% of the subjects had a history of previously suspected dengue infection. This study reflected the proportion of asymptomatic dengue patients requiring better assessment with a serological test.

### Conclusion

The current study highlighted that real cases of dengue infection may be higher than reported, with a high prevalence of dengue seropositivity and a relatively dominant proportion of asymptomatic cases. The study guides physicians to be aware of every dengue infection in tropical countries and prevent the spread of the disease.

1 / 7

**Funding:** This work was supported by Grant from Ministry of Research and Higher Education Technology Indonesia to SM. The funders had no role in study design, data collection and analysis, decision to publish, or preparation of the manuscript.

**Competing interests:** The authors have declared that no competing interests exist.

## Introduction

Dengue is considered a forthcoming disease, with alarming epidemiological patterns for both human health and the global economy [1]. The prevalence of dengue virus (DENV) infection has been predicted to cause 390 million new cases each year and approximately 96 million dengue hemorrhagic fever cases leading to hospitalization each year [2]. Four serotypes of DENV (DENV-1, -2, -3, and -4) have spread rapidly within countries and across regions, causing epidemics and severe dengue disease, hyperendemicity of multiple DENV serotypes in tropical countries, and autochthonous transmission in Europe and the USA [3,4]. However, the incidence of dengue and other infections has been reported to decrease during the pandemic [5,6].

The burden of dengue disease in many dengue-endemic countries, including Indonesia, remains poorly understood because current passive surveillance systems capture only a trivial fraction of all dengue cases. Furthermore, it mostly depends on clinical diagnosis, which excludes milder and atypical disease presentations [2,7]. Dengue transmission in Indonesia, including in Bali, has become hyperendemic and a significant public health problem. Dengue fever has spread throughout Indonesia since the first case was reported in Surabaya in 1968 [8]. All four dengue serotypes have been reported in Indonesian regions, including Western Java, Surabaya, Bali, and Jambi [3,9–13]. The previous study in Bali 2017, DENV-3 was predominant serotype (48%), followed by DENV-1 (28%), DENV-2 (17%), and DENV-4 (4%) [12]. In 2018, the most prevalent serotype was shifted to DENV-2 (48.7%), followed by DENV-3 (36.1%), DENV-1 (9.2%) and DENV-4 (3.4%) [11].

Dengue infection in Bali has become a recurring and significant public health concern. This study aimed to assess the seroprevalence of dengue infection detected using indirect immunoglobulin G (IgG) enzyme-linked immunosorbent assay (ELISA).

## Materials and methods

Cross-sectional seroprevalence surveys were performed between July 2020 and June 2021 among healthy adults and non-dengue patients in Denpasar, Bali. Inclusion criteria was adult subject ($>$ 18 years old) and signed informed consent where the healthy group was defined as the absence of signs of infection and other clinical manifestations of any illness. Subject with any illness was patients admitted to the hospitals with any illnesses and signed the informed consent. Subject was excluded from the study if they have a history of immune deficiency disease such as HIV, or chronic steroid user. Sampling method of the study employed consecutive sampling method.

Numbers of samples needed was determined using the sample size hypothesis test for different proportions, with the first proportion in dengue cases and the second proportion as healthy exposed.

$$n = \frac{\left(z_{1-\alpha/2}\sqrt{2P(1-P)} + z_{1-\beta}\sqrt{vP1(1-P1) + P2(1-P2)}\right)^2}{\left(P_1 - P_2\right)^2} \tag{1}$$

Where Za (1-alpha) with alpha 0.01 = 2.576, Zb (1-beta) with beta 0.05 = 1.96, P1 as case proportion = 70% (the prevalence from previous study in 2018), P2 as non-case proportion = 50% (an assumption). n = ⟦(2.576√(2x0.6x(1−0.6)) +1.96 √(0.7(1−0.7)+0.5(1−0.5))) ⟧ ^2/ ⟦(0.7−0.5)⟧ ^2 = 242.43. This means that the total samples required was 242 sample, minimally.

Blood samples were collected from 539 randomly selected urban sites in Denpasar. DENV IgG antibodies were detected in serum. Each participant obtained a blood sample of 3 mL of venous blood in clot activator tubes. The serum was separated by centrifugation at 1300 ×g for

10 min, kept at +4°C, and transported to the laboratory of Universitas Warmadewa on the days for further processing.

DENV anti-IgG antibody detection in the collected sera was performed using a commercial ELISA (catalog number EI 266a-9601-1 G, Panbio, Luebeck, Germany) following the manufacturer's instructions. The specificity and sensitivity of the kit were 0.988 (95% confidence interval [CI]: 0.979–0.993) and 0.892 (95% CI: 0.879–0.903), respectively. Briefly, serum samples diluted 1:101 were added to microplates, and optical densities were measured using a microplate reader. Antibody values of $\geq$ 20 relative units/mL were considered seropositive.

### Ethical considerations

The study protocol was approved by the Institutional Review Board of the Faculty of Medicine, Udayana University, Bali, Indonesia (approval no. 485/UN.14.2/KEP/2020). All study participants were enrolled after obtaining their written informed consent.

### Statistical analysis

Descriptive analysis was used to describe dengue seroprevalence using an indirect IgG ELISA. The chi-squared test was used to analyze the association between variables.

### Results

This study found a high prevalence of dengue infection among the participants. The seroprevalence in most age groups was higher than 80% (Fig 1). Moreover, the 41-45-year-old-group and older were observed to have a higher positivity rate than the younger group. As a result, in the following analysis, we divided the group into two age groups ($< 40$ and $\geq 40$ years groups). The baseline characteristics of the participants are listed in Table 1. The oldest age of the subject was 86 years old.

The results showed a high prevalence of seropositive dengue among participants (Table 1). The study found 85.5% of the subject was Dengue antibody detected, of which 336 (72.9%) was healthy subject (Table 2). Among febrile subjects, the diagnosis was suspected viral infection other than Dengue diagnosis. The most diagnosis of non-febrile was acute diarrhoea, acute gastritis, diabetes mellitus, as high as 22.2%, 20% and 17,8%, respectively.

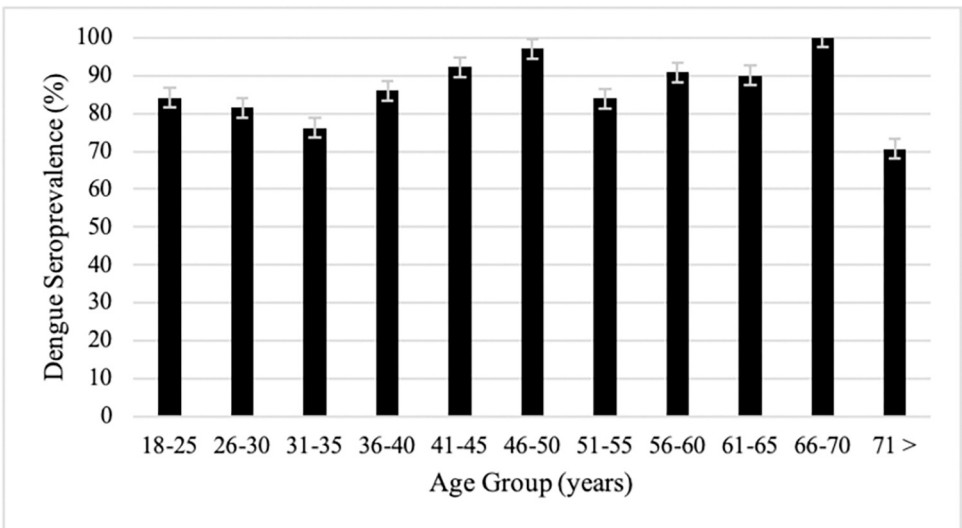

**Fig 1. Dengue antibody seroprevalence by age groups.**

**Table 1. Baseline characteristics of the participants (N = 539).**

| Variables | Frequency | Percentage (%) |
|---|---|---|
| **Age** | | |
| Median (IQR) | 34.1 (18–86.1) | |
| < 40 years | 330 | 61.2 |
| ≥ 40 years | 209 | 38.8 |
| **Sex** | | |
| Male | 294 | 54.5 |
| Female | 245 | 45.5 |
| **Indirect IgG dengue** | | |
| Negative | 72 | 13.4 |
| Equivocal | 6 | 1.1 |
| Positive | 461 | 85.5 |
| **Participants' status** | | |
| Healthy | 373 | 69.2 |
| Any illness | 166 | 37.8 |
| **Diagnosis (N = 166)** | | |
| COVID-19 | 43 | 25.9 |
| Febrile illness | 33 | 19.9 |
| Non-febrile illness | 90 | 54.2 |

COVID-19, coronavirus disease; IgG, immunoglobulin G; IQR, interquartile range.

## Discussion

The burden of dengue infection is not well-documented in Indonesia, particularly in the adult population. This is the first study on dengue seroprevalence among adult populations in Bali, where dengue is endemic. The current study findings, in which dengue seropositivity was as high as 85.5%, revealed that people in tropical countries are at risk of dengue infection. Another seroprevalence study reported a lower rate of dengue infection in Indonesian children (69.4%) [14]. This difference may be associated with a lower probability of *Aedes* bite among children than among adults. Our results are in line with those of another study in Thailand

**Table 2. Association between the variables.**

| Variable | | Indirect IgG ELISA | | | | p | Odds ratio | Confidence interval (95%) |
|---|---|---|---|---|---|---|---|---|
| | | Positive | | Negative | | | | |
| | | N | % | N | % | | | |
| **Age** | | | | | | | | |
| < 40 years | 330 | 271 | 82.1 | 59 | 17.9 | 0.005 | 0.459 | 0.265–0.796 |
| ≥ 40 years | 209 | 190 | 90.9 | 19 | 9.1 | | | |
| **Sex** | | | | | | | | |
| Male | 294 | 252 | 85.7 | 42 | 14.3 | 0.893 | 0.968 | 0.598–1.566 |
| Female | 245 | 209 | 85.3 | 36 | 14.7 | | | |
| **Participants' status** | | | | | | | | |
| Healthy | 373 | 336 | 90.1 | 37 | 9.9 | < 0.001 | 0.336 | 0.206–0.548 |
| Any illness | 166 | 125 | 75.3 | 41 | 24.7 | | | |

IgG, immunoglobulin G; DENV, dengue virus; ELISA, enzyme-linked immunosorbent assay; DHF, dengue hemorrhagic fever.

that reported a high seroprevalence rate (91.5%) in adults and children in urban and rural areas [15]. A study in Dhaka reported a seroprevalence as high as 80% [16]. Carabali et al. [17] reported that the seroprevalence of dengue infection was 61%, and only 3.3% of seropositive subjects had a history of dengue [17]. A relatively small proportion (31.3%) of dengue seropositivity was found in a study in India [18]. The term seropositive dengue consists of IgM or IgG or both IgG, and IgM was positive. It was not clear the seropositivity of dengue based on IgG dengue only such as this study. The discrepancy may be explained by the study in India was conducted in both rural and urban areas, but not in our study provided in the urban area only.

The current study also found that participants aged > 40 years had a significantly higher proportion of dengue seropositivity than participants aged < 40 years (p = 0.005). This finding may be related to the higher risk of *Aedes* bites than those who are younger. This is consistent with another study in Thailand, which found that more than 98% of participants older than 25 years were dengue seropositive [19]. In this study, the healthy group had a higher seroprevalence than any other illness group (90.1% vs 75.3%), with statistical significance (p<0.001). This result was in line with current finding where seropositivity was higher as age advanced. Study showed that as the longer people settle in a dengue-endemic area, risk of exposure were increased coupled with the lifelong persistence of anti-DENV antibodies [20]. Immunodeficiency may affect the production of the antibody and this already been used as an exclusion criterion. However, the assessment of the immune deficiency subject was merely based on the interview without any objective data therefore the ill subject has lower dengue seropositive rate, compared to healthy subjects.

Since the history of suspected dengue infection was previously reported in only 60% of the participants, we predicted that there would be an asymptomatic dengue infection among the participants, compared with the results of the indirect IgG dengue ELISA. The study results reflect the proportion of asymptomatic dengue cases that need to be assessed with a serological study. With this high seroprevalence (85.5%), the population has a higher chance of being infected multiple times with dengue. Multiple exposures to this virus increase the risk of severe disease [21,22]. An individual infected with a different dengue serotype for the second time may experience an antibody-dependent enhancement of infection, which means that antibodies from the previous infection serve to disseminate the viral infection and increase viremia [21]. Even though no focus reduction neutralization test (FRNT) was done to confirm ADE, previous study in Indonesia reported that there were rather vast number of children had enhancing antibodies as measured by modified semi-adherent human erythroleukemia K562 cells assay and conventional neutralization test [23]. Another study also shown that there was enhancing antibody arrangement in contrast to Indonesian strains of DENV-2 and DENV-3 [24]. Therefore, the possibility of having a high number of hospitalized patients with dengue infection should be of concern to the Bali Public Health Office.

The limitation of the study was recall bias related to dengue infection history among the participants. Since we only used indirect IgG ELISA, we were unable to distinguish between primary and secondary infections. On the other hand, the IgG antibodies of the DENV remain on the body for more than 5 years after the infection, and the disease from years prior may have been detected in this study. Indeed, the cross-reactivity among flavivirus infections needs to be determined with the plaque reduction neutralization test study, which is not the case in this study.

## Conclusions

The current study highlighted a high prevalence of dengue seropositive individuals which indicates that a large proportion of the population already infected with dengue may have primary

dengue infection. Given the ADE theory that secondary infections lead to more severe dengue infections, this finding is very useful for physicians in the tropics to be alert to the occurrence of severe dengue infection.

## Supporting information

**S1 Fig. STROBE flowchart.**
(TIF)

## Acknowledgments

We would like to thank all participants, nurses, and the research team at the Faculty of Medicine and Health Sciences, Universitas Warmadewa, for supporting this study.

## Author Contributions

**Conceptualization:** Sri Masyeni.

**Data curation:** A. A. A. L. Paramasatiari, Ananda Maheraditya, Ratna Kartika Dewi.

**Formal analysis:** N. W. Winianti, Agus Santosa, Nyoman Trisna Sumadewi.

**Funding acquisition:** Sri Masyeni.

**Investigation:** Marta Setiabudy, Sianny Herawati.

**Writing – original draft:** Sri Masyeni, Rois Muqsith Fatawy.

**Writing – review & editing:** Rois Muqsith Fatawy.

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
