## [Decision Letter · Decision Letter 0]

9 Sep 2022

PONE-D-22-19419Dengue seroprevalence study during COVID-19 pandemic in BaliPLOS ONE

Dear Dr. Masyeni,

Thank you for submitting your manuscript to PLOS ONE. After careful consideration, we feel that it has merit but does not fully meet PLOS ONE’s publication criteria as it currently stands. Therefore, we invite you to submit a revised version of the manuscript that addresses the points raised during the review process.

As you can see from the report attached below, your manuscript has been evaluated by a single reviewer who has carried out a thorough assessment and has provided constructive feedback to strengthen your study. They raised several concerns about the study design and methodological reporting, and I encourage you to pay particular attention to addressing these aspects as they affect the overall validity of your findings and conclusions. Please also note that even though we have only been able to secure a single reviewer to assess your manuscript, we are issuing a decision at this point to prevent further delays. Please be aware that the editor who handles your revised manuscript might find it necessary to invite additional reviewers to assess this work once the revised manuscript is submitted. However, we will aim to proceed on the basis of this single review if possible.    

We look forward to receiving your revised manuscript.

Kind regards,

Dario Ummarino, PhD

Senior Editor

PLOS ONE

Journal Requirements:

2. PLOS requires an ORCID iD for the corresponding author in Editorial Manager on papers submitted after December 6th, 2016. Please ensure that you have an ORCID iD and that it is validated in Editorial Manager. To do this, go to ‘Update my Information’ (in the upper left-hand corner of the main menu), and click on the Fetch/Validate link next to the ORCID field. This will take you to the ORCID site and allow you to create a new iD or authenticate a pre-existing iD in Editorial Manager. Please see the following video for instructions on linking an ORCID iD to your Editorial Manager account: https://www.youtube.com/watch?v=_xcclfuvtxQ.

Reviewers' comments:

Reviewer's Responses to Questions

**Comments to the Author**

1. Is the manuscript technically sound, and do the data support the conclusions?

Reviewer #1: Partly

2. Has the statistical analysis been performed appropriately and rigorously? 

Reviewer #1: Yes

3. Have the authors made all data underlying the findings in their manuscript fully available?

Reviewer #1: Yes

4. Is the manuscript presented in an intelligible fashion and written in standard English?

Reviewer #1: No

5. Review Comments to the Author

Reviewer #1: The authors have been concise, which is to be appreciated, but at the expense of explaining how the surveys were done, and that will affect the conclusions that might be reached.

Major points

1) I suggest the authors use the STROBE checklist for reporting of observational studies in epidemiology (von Elm et al BMJ 2007). Please include the checklist in any future version.

2) The abstract says: “clinical manifestation of dengue varies from asymptomatic cases to severe dengue manifestation. The detection of clinical cases enables us to measure the incidence of dengue infection”. This is contradictory. Since some cases are asymptomatic, detection of clinical cases will, in fact, underestimate the incidence of infection.

3) Then the last part of the same sentence is also misleading: “whereas serological surveys give insights into the prevalence of infection.” In fact, serological surveys (as in the current study) are generally for long-lasting antibodies, so are informative about the experience of past infection, not prevalence of current infection, as seems to be suggested. Also, in context, the statement suggests that serosurveys don’t give information about incidence of infection. However, repeat surveys in the same people can do so.

4) Then the final sentence of the introduction of the abstract is “This study aimed to determine the dengue prevalence among healthy adult patients in Bali.” Given the previous exposition, it’s not clear what’s meant by “prevalence” (i.e. prevalence of what?). Also, how can they be both healthy and patients? (See below about doubts about the population.)

5) In the results section of the abstract, measures of association (e.g. odds ratios) and confidence intervals should be included, not only p values.

6) In the concluding section of the abstract says: “The current study highlighted a high prevalence of dengue seropositive with a relatively dominant proportion of asymptomatic cases.” The second part is not worth highlighting because the participants are said to be healthy. Although the exact population is not really clear (see comments about methods below).

7) The English should be reviewed by a native speaker. Generally it is comprehensible but there are exceptions e.g. the first sentence of the introduction: “Dengue has been considered the disease of the forthcoming”.

8) In the introduction, again there is apparent confusion between epidemiological terms e.g. “The prevalence of dengue virus (DENV) infection has been predicted to cause 390 million new cases arising each year”. And the introduction closes by saying “The study objective is to assess the prevalence of dengue infection detected by Indirect IgG ELISA” which is misleading because IgG is about history of infection, not only recent or current infections. The same issue arises on line 84 of the results: please check throughout.

9) The methods section starts as follows. “Cross-sectional seroprevalence surveys performed in July 2020 – June 2021 among healthy adults and non-dengue patients in Denpasar Bali. Blood samples collected from each of 539 randomly selected samples from urban sites in Denpasar.” This raises several important questions. How many surveys were done? What was the random sampling mechanism? What were the inclusion criteria? Were some or all of the people included more than once? If the aim was to sample healthy people in their houses then why specify non-dengue patients, and how is it that more than a third of the people (per Table 1) were not healthy? E.g. is this illness at the time of the survey, or history of illness? Line 110, in the discussion, mentions history of these participants but it was not mentioned before.

10) In the analysis, I suggest plotting the seroprevalence by age, which will give some idea of the force of infection.

11) Line 96-97: “The current dengue seroprevalence study conducted on adults in Bali reveals that people in tropical countries are at risk of dengue infection”. There should be a more nuanced discussion of the generalizability of the findings.

12) Line 105-107: “The current study also found that participants aged more than 40 years old had a significantly higher proportion of dengue seropositive than participants aged less than 40 (p=0.005). This finding may be related to the risk of Aedes bites being higher than younger age in their life.” The finding of increased seroprevalence with age is completely expected, and the most economical explanation is simply that older people have had greater time of exposure.

13) COVID is mentioned in the title but this isn’t prominent in the interpretation. If the pandemic context is important I suggest explaining why and making relevant conclusions; if it’s not important then it doesn’t need to be mentioned in the title.

Minor points

14) The results section of the abstract says “Another seroprevalence study reported a lower rate of dengue infection in children in Indonesia”. First, if this is a separate study then it should not be reported in the results section of the abstract of the current one. Then, again, it’s at best confusing to talk about “dengue infection” here, because it was presumably not current infection. It would be better to talk about “proportion seropositive”.

15) “Aedes” should be in italics throughout.

16) Line 70-71. “The kit’s specificity and sensitivity are 0.988 (95% CI: 0.979–0.993) and 0.892 (95% CI: 0.879–0.903), respectively.” I suppose this is from a previous source, which should then be cited.

17) Table 2. Correct spelling: “positif” and “negatif”.

18) Line 100-102: “This study result is in line with another study that report the seroprevalence rate was high (91.5%) across all eight study sites. The reason is the study was conducted on adults and children in urban and rural populations in Thailand”. The ordering is not clear (eight study sites are mentioned before the study country) and stated reason isn’t very explanatory. Overall, I think the message from this section should be that the seroprevalence is high in the current study, but in line with comparable studies from the region and beyond.

6. PLOS authors have the option to publish the peer review history of their article (what does this mean?). If published, this will include your full peer review and any attached files.

Reviewer #1: No

---

## [Author Response · Author response to Decision Letter 0]

11 Nov 2022

Dear Honored editor and reviewer,

Please find our answers to each of your comments below.

Thank you so much for your valuable time.

Best regards,

Authors

Reviewer #1: The authors have been concise, which is to be appreciated, but at the expense of explaining how the surveys were done, and that will affect the conclusions that might be reached.

Major points

1) I suggest the authors use the STROBE checklist for reporting of observational studies in epidemiology (von Elm et al BMJ 2007). Please include the checklist in any future version.

Answer: Thank you for reminding us. We have already checked using the STROBE checklist and added any information needed.

2) The abstract says: “clinical manifestation of dengue varies from asymptomatic cases to severe dengue manifestation. The detection of clinical cases enables us to measure the incidence of dengue infection”. This is contradictory. Since some cases are asymptomatic, detection of clinical cases will, in fact, underestimate the incidence of infection.

Answer: Thank you so much for addressing this. We have already rewritten the sentence and made it clearer.

3) Then the last part of the same sentence is also misleading: “whereas serological surveys give insights into the prevalence of infection.” In fact, serological surveys (as in the current study) are generally for long-lasting antibodies, so are informative about the experience of past infection, not prevalence of current infection, as seems to be suggested. Also, in context, the statement suggests that serosurveys don’t give information about incidence of infection. However, repeat surveys in the same people can do so.

Answer: Thank you so much. We have already rewritten the sentence

4) Then the final sentence of the introduction of the abstract is “This study aimed to determine the dengue prevalence among healthy adult patients in Bali.” Given the previous exposition, it’s not clear what’s meant by “prevalence” (i.e. prevalence of what?). Also, how can they be both healthy and patients? (See below about doubts about the population.)

Answer: Thank you so much. We have already clarified and rewritten the sentence.

5) In the results section of the abstract, measures of association (e.g. odds ratios) and confidence intervals should be included, not only p values.

Answer: We have already added the odds ratios and confidence intervals in the abstracts.

6) In the concluding section of the abstract says: “The current study highlighted a high prevalence of dengue seropositive with a relatively dominant proportion of asymptomatic cases.” The second part is not worth highlighting because the participants are said to be healthy. Although the exact population is not really clear (see comments about methods below).

Answer: We have clarified the terms we used to describe the population in the current manuscript. 

7) The English should be reviewed by a native speaker. Generally it is comprehensible but there are exceptions e.g. the first sentence of the introduction: “Dengue has been considered the disease of the forthcoming”.

Answer: The current revision has already been proofread by English Editing Service editage.com.

8) In the introduction, again there is apparent confusion between epidemiological terms e.g. “The prevalence of dengue virus (DENV) infection has been predicted to cause 390 million new cases arising each year”. And the introduction closes by saying “The study objective is to assess the prevalence of dengue infection detected by Indirect IgG ELISA” which is misleading because IgG is about history of infection, not only recent or current infections. The same issue arises on line 84 of the results: please check throughout.

Answer: Thank you so much. We already revised the objective statement.

9) The methods section starts as follows. “Cross-sectional seroprevalence surveys performed in July 2020 – June 2021 among healthy adults and non-dengue patients in Denpasar Bali. Blood samples collected from each of 539 randomly selected samples from urban sites in Denpasar.” This raises several important questions. 

How many surveys were done? Once

What was the random sampling mechanism? Collecting sample was done by consecutive sampling techniques

What were the inclusion criteria? We have already added it in the method section.

Were some or all of the people included more than once? No.

If the aim was to sample healthy people in their houses then why specify non-dengue patients, and how is it that more than a third of the people (per Table 1) were not healthy? 

Thank you so much for raising this issue. Our participants are both adult healthy people and patients. The patients itself comprised of COVID-19, febrile illness and non-febrile patients. 

 E.g. is this illness at the time of the survey, or history of illness? Illness at the time of the survey

 Line 110, in the discussion, mentions history of these participants but it was not mentioned before.

Answer: We have added the results

10) In the analysis, I suggest plotting the seroprevalence by age, which will give some idea of the force of infection.

Answer: Thank you we add a graph explaining seroprevalence by age group (@5 years) 

11) Line 96-97: “The current dengue seroprevalence study conducted on adults in Bali reveals that people in tropical countries are at risk of dengue infection”. There should be a more nuanced discussion of the generalizability of the findings.

Answer: We have already added some of the discussion.

12) Line 105-107: “The current study also found that participants aged more than 40 years old had a significantly higher proportion of dengue seropositive than participants aged less than 40 (p=0.005). This finding may be related to the risk of Aedes bites being higher than younger age in their life.” The finding of increased seroprevalence with age is completely expected, and the most economical explanation is simply that older people have had the greater time of exposure.

Answer: Thank you for clarifying

13) COVID is mentioned in the title but this isn’t prominent in the interpretation. If the pandemic context is important I suggest explaining why and making relevant conclusions; if it’s not important then it doesn’t need to be mentioned in the title.

Answer: We revised the title into dengue seroprevalence study in Bali

Minor points

14) The results section of the abstract says “Another seroprevalence study reported a lower rate of dengue infection in children in Indonesia”. First, if this is a separate study then it should not be reported in the results section of the abstract of the current one. Then, again, it’s at best confusing to talk about “dengue infection” here, because it was presumably not current infection. It would be better to talk about “proportion seropositive”.

Answer: Thank you for your input. We have already deleted and rewritten the sentence in the abstract.

15) “Aedes” should be in italics throughout.

Answer: Revised

16) Line 70-71. “The kit’s specificity and sensitivity are 0.988 (95% CI: 0.979–0.993) and 0.892 (95% CI: 0.879–0.903), respectively.” I suppose this is from a previous source, which should then be cited.

Answer: Revised

17) Table 2. Correct spelling: “positif” and “negatif”.

Answer: Revised

18) Line 100-102: “This study result is in line with another study that report the seroprevalence rate was high (91.5%) across all eight study sites. The reason is the study was conducted on adults and children in urban and rural populations in Thailand”. The ordering is not clear (eight study sites are mentioned before the study country) and stated reason isn’t very explanatory. Overall, I think the message from this section should be that the seroprevalence is high in the current study, but in line with comparable studies from the region and beyond.

Answer: Thank you for your input. We have already rewritten the sentence.

---

## [Decision Letter · Decision Letter 1]

16 Jan 2023

PONE-D-22-19419R1Dengue seroprevalence study in BaliPLOS ONE

Dear Dr. Masyeni,

Thank you for submitting your manuscript to PLOS ONE. After careful consideration, we feel that it has merit but does not fully meet PLOS ONE’s publication criteria as it currently stands. Therefore, we invite you to submit a revised version of the manuscript that addresses the points raised during the review process.

We look forward to receiving your revised manuscript.

Kind regards,

Kovy Arteaga-Livias

Academic Editor

PLOS ONE

Reviewers' comments:

Reviewer's Responses to Questions

**Comments to the Author**

1. If the authors have adequately addressed your comments raised in a previous round of review and you feel that this manuscript is now acceptable for publication, you may indicate that here to bypass the “Comments to the Author” section, enter your conflict of interest statement in the “Confidential to Editor” section, and submit your "Accept" recommendation.

Reviewer #2: All comments have been addressed

Reviewer #3: (No Response)

2. Is the manuscript technically sound, and do the data support the conclusions?

Reviewer #2: Partly

Reviewer #3: Partly

3. Has the statistical analysis been performed appropriately and rigorously? 

Reviewer #2: N/A

Reviewer #3: Yes

4. Have the authors made all data underlying the findings in their manuscript fully available?

Reviewer #2: Yes

Reviewer #3: Yes

5. Is the manuscript presented in an intelligible fashion and written in standard English?

Reviewer #2: Yes

Reviewer #3: Yes

6. Review Comments to the Author

Reviewer #2: - In line 44 and 45, authors described that “Four serotypes of DENV (DENV- 1, -2, -3, and -4) have spread rapidly within countries and across regions, causing epidemics” in line 44 and 45.

Could authors describe the prevalence of the serotype of DENV in Bali?

- Authors described that “The study found 85.5% of the subject was Dengue antibody detected, of which 336 (72.9%) was healthy subject (Table 2)” between line 105 and 107.

I am understanding that most people already experience infection of dengue virus and the authors also mentioned ADE (antibody-dependent enhancement) of infection (between line 142 and 144).

Did authors check FRNT50 in this study?

Because after first infection of dengue virus, non-neutralizing antibodies to a specificity serotype of dengue virus are produced and these antibodies are harmful such as shock and hemorrhage after others serotypes are infected (authors also mentioned below.

An individual infected with a different dengue serotype for the second time may experience an antibody-dependent enhancement of infection, which means that antibodies from the previous infection serve to disseminate the viral infection and increase viremia [20])

Therefore, I recommend that authors also showed result of FRNT50 in this study.

- In line 121, authors described that “This difference may be associated with a lower probability of Aedes bite among children than among adults” and “This finding may be related to the higher risk of Aedes bites than those who are younger” in line132 and 133.

Could you put the reason in this part?

Because I think that the chance of a bite maybe same (or similarly) between children and adults.

Reviewer #3: Manuscript “Dengue seroprevalence study in Bali”

This manuscript by Masyeni et al. reports a study of dengue seroprevalence in Bali. It employed a cross-sectional seroprevalence survey methodology among healthy and ill adult patients in Denpasar, Bali. The authors noted the high prevalence of dengue seropositivity in Bali. I do have some comments/suggestions that I think would help to strengthen the manuscript.

Major comments/suggestions

1. The objective of study (lines 57-59); this research conducted a seroprevalence survey of dengue infection rather than assessing the prevalence of dengue infection, because single IgG antibody indicates a previous dengue infection rather than an acute or asymptomatic dengue infection. Please revise the text to” “to assess the seroprevalence of dengue infection….”

2. The authors should detail sample-size calculations, including the healthy, febrile, and non-febrile subgroup.

3. The authors should provide the study’s eligibility (inclusion/exclusion) criteria.

4. How did the authors randomize participants in the healthy, febrile, and non-febrile groups for enrollment? Please clarify the methodology in the manuscript.

5. The authors should provide a study flow chart in the manuscript.

6. The authors should provide additional relevant data, including educational level, occupation, type of household, behavior risks, knowledge of dengue, etc.

7. Table 2; the seroprevalence in the healthy group was higher than any other illness group (90.1% vs 75.3%) with statistically significance; the authors should add some discussion for this finding.

8. Conclusions, lines 154-156. “The current study highlighted a high prevalence of dengue seropositive individuals with a relatively high proportion of asymptomatic dengue cases. The study results can guide physicians on the risk of more severe dengue in every dengue infection in tropical countries.” Actually, this study was unable to detect asymptomatic dengue cases, because a single IgG antibody cannot indicate acute dengue infection. Furthermore, the study was not designed to detect risk/associated factors for severe dengue. Please revise the conclusions accordingly.

Minor comments/suggestions

1. Line 106-107; “….of which 336 (72.9%) was healthy subject (Table 2).” In table 2, 336 participants (90.1%) were shown as being in the healthy group, so what does the 72.9% refer to?

2. In response to the reviewer’s comments, the authors should inform us what line numberings have been changed in the revised manuscript.

7. PLOS authors have the option to publish the peer review history of their article (what does this mean?). If published, this will include your full peer review and any attached files.

Reviewer #2: No

Reviewer #3: No

---

## [Author Response · Author response to Decision Letter 1]

4 Mar 2023

Dear Academic Editor and Reviewer,

We would like to thank you for reviewing our articles. Please find our answer to the concern raised below.

Thank you.

Best regards,

Authors

Response to the Reviewer

Reviewer #2: - In lines 44 and 45, authors described that “Four serotypes of DENV (DENV- 1, -2, -3, and -4) have spread rapidly within countries and across regions, causing epidemics” in line 44 and 45.

Could authors describe the prevalence of the serotype of DENV in Bali?

Our Response: Thank you. We have described each prevalence of the serotype of DENV in the introduction. (line 56-59)

- Authors described that “The study found 85.5% of the subject was Dengue antibody detected, of which 336 (72.9%) was healthy subject (Table 2)” between line 105 and 107.

I am understanding that most people already experience infection of dengue virus and the authors also mentioned ADE (antibody-dependent enhancement) of infection (between line 142 and 144).

Did authors check FRNT50 in this study?

Our Response: The study was conducted during the pandemic, we are short in budget and lab resources, so we did not conduct FRNT50. 

Because after first infection of dengue virus, non-neutralizing antibodies to a specificity serotype of dengue virus are produced and these antibodies are harmful such as shock and hemorrhage after others serotypes are infected (authors also mentioned below.

An individual infected with a different dengue serotype for the second time may experience an antibody-dependent enhancement of infection, which means that antibodies from the previous infection serve to disseminate the viral infection and increase viremia [20])

Therefore, I recommend that authors also showed result of FRNT50 in this study.

Our Response: As we mentioned, the study was conducted during pandemic, we are short in budget and lab resources, so we did not conduct either FRNT or PRNT. We have also mentioned this as a study limitation in the discussion.

- In line 121, authors described that “This difference may be associated with a lower probability of Aedes bite among children than among adults” and “This finding may be related to the higher risk of Aedes bites than those who are younger” in line132 and 133.

Could you put the reason in this part?

Because I think that the chance of a bite maybe same (or similarly) between children and adults.

Our Response: The likelihood of infection increases with age. Even in small concentrations, neutralizing antibodies to the same serotype can still be detected.

Reviewer #3: Manuscript “Dengue seroprevalence study in Bali”

This manuscript by Masyeni et al. reports a study of dengue seroprevalence in Bali. It employed a cross-sectional seroprevalence survey methodology among healthy and ill adult patients in Denpasar, Bali. The authors noted the high prevalence of dengue seropositivity in Bali. I do have some comments/suggestions that I think would help to strengthen the manuscript.

Major comments/suggestions

1. The objective of study (lines 57-59); this research conducted a seroprevalence survey of dengue infection rather than assessing the prevalence of dengue infection, because single IgG antibody indicates a previous dengue infection rather than an acute or asymptomatic dengue infection. Please revise the text to” “to assess the seroprevalence of dengue infection….”

Our response: Thank you for your suggestion. We have revised the study objective. (line 61)

2. The authors should detail sample-size calculations, including the healthy, febrile, and non-febrile subgroup.

Our response: We determine the minimum sample size for our cross-sectional study using the sample size hypothesis test for different proportions, with the first proportion in dengue cases and the second proportion as healthy exposed.

Where

Za (1-alfa) with alpha 0.01= 2.576

Zb (1-beta) with beta 0.05= 1.96

P1 as case proportion = 70% (the prevalence from previous study in 2018)

P2 as non-case proportion = 50% (an assumption)

p̂= (P1+P2)/2 = 0.6

n=〖(2.576√(2x0.6x(1-0.6))+1.96 √(0.7(1-0.7)+0.5(1-0.5)))〗^2/〖(0.7-0.5)〗^2

n= 9.697/0.04 = 242.43

3. The authors should provide the study’s eligibility (inclusion/exclusion) criteria.

Our response: Thank you so much. We have already written the study’s eligibility in the method. (line 66)

4. How did the authors randomize participants in the healthy, febrile, and non-febrile groups for enrollment? Please clarify the methodology in the manuscript.

Our response: The sampling method used snowball/consecutive random sampling.

5. The authors should provide a study flow chart in the manuscript.

Our response: Thanks for reminding. We have already added as a supplementary figure.

6. The authors should provide additional relevant data, including educational level, occupation, type of household, behavior risks, knowledge of dengue, etc.

Our response: Thank you for suggesting. However, the study did not address the issues.

7. Table 2; the seroprevalence in the healthy group was higher than any other illness group (90.1% vs 75.3%) with statistically significance; the authors should add some discussion for this finding.

Our response: Thank you for reminding us. We have already added the discussion about this. (line 143)

8. Conclusions, lines 154-156. “The current study highlighted a high prevalence of dengue seropositive individuals with a relatively high proportion of asymptomatic dengue cases. The study results can guide physicians on the risk of more severe dengue in every dengue infection in tropical countries.” Actually, this study was unable to detect asymptomatic dengue cases, because a single IgG antibody cannot indicate acute dengue infection. Furthermore, the study was not designed to detect risk/associated factors for severe dengue. Please revise the conclusions accordingly.

Our response: Thank you so much. We have already revised the conclusion of this study (line 167)

Minor comments/suggestions

1. Line 106-107; “….of which 336 (72.9%) was healthy subject (Table 2).” In table 2, 336 participants (90.1%) were shown as being in the healthy group, so what does the 72.9% refer to?

Our response: 72.9% refer to the proportion between 336 positive participants of healthy group per total positive participants 461(85.5%) from both group healthy and any illness.

2. In response to the reviewer’s comments, the authors should inform us what line numberings have been changed in the revised manuscript.

Our response: We have added line numberings in this letter.

---

## [Decision Letter · Decision Letter 2]

13 Apr 2023

PONE-D-22-19419R2Dengue seroprevalence study in BaliPLOS ONE

Dear Dr. Masyeni,

Thank you for submitting your manuscript to PLOS ONE. After careful consideration, we feel that it has merit but does not fully meet PLOS ONE’s publication criteria as it currently stands. Therefore, we invite you to submit a revised version of the manuscript that addresses the points raised during the review process.

We look forward to receiving your revised manuscript.

Kind regards,

Kovy Arteaga-Livias

Academic Editor

PLOS ONE

Reviewers' comments:

Reviewer's Responses to Questions

**Comments to the Author**

1. If the authors have adequately addressed your comments raised in a previous round of review and you feel that this manuscript is now acceptable for publication, you may indicate that here to bypass the “Comments to the Author” section, enter your conflict of interest statement in the “Confidential to Editor” section, and submit your "Accept" recommendation.

Reviewer #2: All comments have been addressed

Reviewer #3: All comments have been addressed

2. Is the manuscript technically sound, and do the data support the conclusions?

Reviewer #2: Yes

Reviewer #3: Yes

3. Has the statistical analysis been performed appropriately and rigorously? 

Reviewer #2: Yes

Reviewer #3: Yes

4. Have the authors made all data underlying the findings in their manuscript fully available?

Reviewer #2: Yes

Reviewer #3: Yes

5. Is the manuscript presented in an intelligible fashion and written in standard English?

Reviewer #2: Yes

Reviewer #3: Yes

6. Review Comments to the Author

Reviewer #2: - Authors described that “The study found 85.5% of the subject was Dengue antibody detected, of which 336 (72.9%) was healthy subject (Table 2)” between line 105 and 107.

I am understanding that most people already experience infection of dengue virus and the authors also mentioned ADE (antibody-dependent enhancement) of infection (between line 142 and 144).

Did authors check FRNT50 in this study?

Our Response: The study was conducted during the pandemic, we are short in budget and lab resources, so we did not conduct FRNT50.

: Could authors put reference (s) about ADE in this region if authors had no data about ADE? Because the ADE issue is very important in Dengue virus infection.

Reviewer #3: Manuscript “Dengue seroprevalence study in Bali”

According to their responses and the revised manuscript, the authors appear to have addressed the reviewers' comments. However, the manuscript still requires minor revisions before acceptance for publication:

1. The authors should demonstrate their sample-size calculations for the healthy and any illness subgroups.

2. The sample-size calculations should be included in the manuscript.

3. Lines 143-148: discussion of the statistically significantly higher seroprevalence among the healthy group compared with the any illness group needs strengthening. Individuals with a history of immunodeficiency were excluded according to the exclusion criteria. Therefore, individuals with obvious/severe immunodeficiency conditions should not be enrolled into this study. However, if mildly immunodeficient individuals were unintentionally enrolled into the study, they should be represented in both subgroups. Therefore, the authors’ discussion may not make sense. The authors may look for other factors such as age of the participants in both subgroups because seroprevalence of dengue increase according to their older age.

7. PLOS authors have the option to publish the peer review history of their article (what does this mean?). If published, this will include your full peer review and any attached files.

Reviewer #2: No

Reviewer #3: No

---

## [Author Response · Author response to Decision Letter 2]

19 May 2023

Response to the Reviewer

Reviewer #2: - Authors described that “The study found 85.5% of the subject was Dengue antibody detected, of which 336 (72.9%) was healthy subject (Table 2)” between line 105 and 107.

I am understanding that most people already experience infection of dengue virus and the authors also mentioned ADE (antibody-dependent enhancement) of infection (between line 142 and 144).

Did authors check FRNT50 in this study?

Our Response: The study was conducted during the pandemic, we are short in budget and lab resources, so we did not conduct FRNT50.

: Could authors put reference (s) about ADE in this region if authors had no data about ADE? Because the ADE issue is very important in Dengue virus infection

Our Response: We added discussion on line 165-170 about ADE in Indonesia, because until now there isn’t any study that asses dengue ADE in Bali, and subsequent references can be found on line 253-260.

Reviewer #3: Manuscript “Dengue seroprevalence study in Bali”

According to their responses and the revised manuscript, the authors appear to have addressed the reviewers' comments. However, the manuscript still requires minor revisions before acceptance for publication:

1. The authors should demonstrate their sample-size calculations for the healthy and any illness subgroups.

2. The sample-size calculations should be included in the manuscript.

3. Lines 143-148: discussion of the statistically significantly higher seroprevalence among the healthy group compared with the any illness group needs strengthening. Individuals with a history of immunodeficiency were excluded according to the exclusion criteria. Therefore, individuals with obvious/severe immunodeficiency conditions should not be enrolled into this study. However, if mildly immunodeficient individuals were unintentionally enrolled into the study, they should be represented in both subgroups. Therefore, the authors’ discussion may not make sense. The authors may look for other factors such as age of the participants in both subgroups because seroprevalence of dengue increase according to their older age.

Our Response:

1&2. We added sample-size calculation as well as the formula that were used to determined minimal number of samples required in the study. This information can be found on line 73-80.

3. Thank you for your suggestion, based on your input we added new explanation on line 152-155 with references on line 250-252

---

## [Decision Letter · Decision Letter 3]

2 Jul 2023

Dengue seroprevalence study in Bali

PONE-D-22-19419R3

Dear Dr. Masyeni,

We’re pleased to inform you that your manuscript has been judged scientifically suitable for publication and will be formally accepted for publication once it meets all outstanding technical requirements.

Kind regards,

Kovy Arteaga-Livias

Academic Editor

PLOS ONE

Additional Editor Comments (optional):

Reviewers' comments:

Reviewer's Responses to Questions

**Comments to the Author**

1. If the authors have adequately addressed your comments raised in a previous round of review and you feel that this manuscript is now acceptable for publication, you may indicate that here to bypass the “Comments to the Author” section, enter your conflict of interest statement in the “Confidential to Editor” section, and submit your "Accept" recommendation.

Reviewer #2: All comments have been addressed

Reviewer #3: All comments have been addressed

2. Is the manuscript technically sound, and do the data support the conclusions?

Reviewer #2: Yes

Reviewer #3: Yes

3. Has the statistical analysis been performed appropriately and rigorously? 

Reviewer #2: Yes

Reviewer #3: Yes

4. Have the authors made all data underlying the findings in their manuscript fully available?

Reviewer #2: Yes

Reviewer #3: Yes

5. Is the manuscript presented in an intelligible fashion and written in standard English?

Reviewer #2: Yes

Reviewer #3: Yes

6. Review Comments to the Author

Reviewer #2: The authors have satisfactorily responded to the reviewer comments. The revised version of manuscript is very nicely improved and the reviewer has no further comments.

Reviewer #3: (No Response)

7. PLOS authors have the option to publish the peer review history of their article (what does this mean?). If published, this will include your full peer review and any attached files.

Reviewer #2: No

Reviewer #3: No

---

## [Editor Report · Acceptance letter]

7 Jul 2023

PONE-D-22-19419R3 

Dengue seroprevalence study in Bali 

Dear Dr. Masyeni:

I'm pleased to inform you that your manuscript has been deemed suitable for publication in PLOS ONE. Congratulations! Your manuscript is now with our production department. 

Kind regards, 

on behalf of

Dr. Kovy Arteaga-Livias 

Academic Editor

PLOS ONE